# Kinematic Assessment of Fine Motor Skills in Children: Comparison of a Kinematic Approach and a Standardized Test

**Ewa Niechwiej-Szwedo** [1,*]**, Taylor A. Brin** [2]**, Benjamin Thompson** [2,3,4] and **Lisa W. T. Christian** [2]

1   Department of Kinesiology and Health Sciences, University of Waterloo, Waterloo, ON N2L 3G1, Canada
2   School of Optometry and Vision Science, University of Waterloo, Waterloo, ON N2L 3G1, Canada;
    ta2brin@uwaterloo.ca (T.A.B.); ben.thompson@uwaterloo.ca (B.T.); lisa.christian@uwaterloo.ca (L.W.T.C.)
3   Centre for Eye and Vision Research, 17W Science Park, Hong Kong
4   Liggins Institute, University of Auckland, Auckland 1010, New Zealand
*   Correspondence: eniechwi@uwaterloo.ca

**Abstract:** Deficits in fine motor skills have been reported in some children with neurodevelopmental disorders such as amblyopia or strabismus. Therefore, monitoring the development of motor skills and any potential improvement due to therapy is an important clinical goal. The aim of this study was to test the feasibility of performing a kinematic assessment within an optometric setting using inexpensive, portable, off-the-shelf equipment. The study also assessed whether kinematic data could enhance the information provided by a routine motor function screening test (the Movement Assessment Battery for Children, MABC). Using the MABC-2, upper limb dexterity was measured in a cohort of 47 typically developing children (7–15 years old), and the Leap motion capture system was used to record hand kinematics while children performed a bead-threading task. Two children with a history of amblyopia were also tested to explore the utility of a kinematic assessment in a clinical population. For the typically developing children, visual acuity and stereoacuity were within the normal range; however, the average standardized MABC-2 scores were lower than published norms. Comparing MABC-2 and kinematic measures in the two children with amblyopia revealed that both assessments provide convergent results and revealed deficits in fine motor control. In conclusion, kinematic assessment can augment standardized tests of fine motor skills in an optometric setting and may be useful for measuring visuomotor function and monitoring treatment outcomes in children with binocular vision anomalies.

**Keywords:** development; vision; amblyopia; visuomotor coordination; visuomotor control; prehension; reaching and grasping





## 1. Introduction

Vision provides a key sensory input when children engage in daily activities such as academics or sports. Visual impairments due to amblyopia and/or strabismus have been associated with real-world visuomotor deficits, including slower and less accurate reaching and grasping movements [1,2], poorer balance and gait [3], reduced reading speed [4,5], and a longer time to fill out scantron cards [6]. Therefore, motor skill improvement should be an important goal of therapies for pediatric binocular vision disorders [7]. However, the clinical assessment of fine motor skills using standardized tests has several limitations: test administration is time-consuming; scoring is based on an observation of task performance, which may introduce bias; there may be a ceiling effect; and the overall score provides limited insight into the nature of the specific underlying visuomotor problem. New technologies have emerged that have the potential to transform and modernize the clinical assessment of fine motor skills by providing objective and detailed information about visuomotor control in addition to global performance metrics.

There are two standardized clinical tools commonly used to assess motor proficiency in children: the Movement Assessment Battery for Children—Second Edition (MABC-2) [8]—and the Bruininks–Oseretsky Test of Motor Proficiency–2 (BOT-2) [9]. Both tests contain multiple tasks that evaluate fine and gross motor skills. Motor impairments are diagnosed when the child's composite score falls below a certain criterion. It is important to use a standardized approach for diagnostic purposes; however, current assessments fail to identify the underlying disruption in visuomotor control that could explain poor performance. For example, a low score on the MABC-2 does not explain why the child is not performing well [10,11], which may be attributed to a disruption in several different sensorimotor processes, such as difficulties in perceiving and selecting the relevant sensory input (i.e., perception and attention); multisensory integration; coordination between task components such as reaching, grasping and placing objects; or deficits in movement planning or execution. Using a high-speed motion camera to record arm and hand movements (i.e., kinematics) provides a highly detailed and objective measurement of visuomotor function [12,13], which might complement the standardized clinical assessment and support the monitoring of treatment effects on real-world behaviours. Notably, previous studies have revealed compensatory kinematic strategies, which would not be evident during a standardized clinical motor test [14]. Understanding sensorimotor adaptions during early development in children is important because maladaptive compensations may lead to difficulties in visuomotor skills at a later stage in development. Thus, the detailed monitoring of motor skill maturation in children with vision impairment could identify children at risk of poor visuomotor development and facilitate an early targeted intervention. To date, detailed kinematic assessments have only been possible in research settings because motion capture systems have been expensive and bulky (i.e., not portable). However, new inexpensive and compact technology is now available that has the potential to enable in-clinic kinematic assessment. The current study examined the utility of a portable motion capture device in assessing fine motor skills in a cohort of children with normal vision and two children with a visual impairment and history of amblyopia. The results from a kinematic assessment were compared to a fine-motor-skills clinical test to determine the feasibility of using the technology-based approach and to investigate whether the kinematic data could augment the standardized clinical motor skill assessment.

## 2. Materials and Methods

### 2.1. Participants

The cohort included 47 typically developing school-aged children between 7 to 15 years old (21 girls, 26 boys; mean age = 10.70 years, SD = 2.35). Children were recruited from the University of Waterloo Optometry Clinic (sample of convenience). Six of the forty-seven children were left-hand dominant, which was determined through self- and parental reports of which hand was preferred for writing and drawing. Children with a medical diagnosis of ADHD, learning/intellectual disability, or autism were excluded from the study. Additionally, two children with a history of amblyopia were assessed (see Table 1 for clinical characteristics).

**Table 1.** Patient characteristics.

|  | **Patient 1** | **Patient 2** |
|---|---|---|
| Age (years); Sex | 12; female | 13; male |
| Type of amblyopia | Strabismic | * NA |
| Visual acuity (logMAR): | Right eye: 0.10 | Right eye: 0.40 |
|  | Left eye: 0.40 | Left eye: 0.70 |
| Stereoacuity (arc sec) | >400 | >400 |
| Handedness | Left | Right |

* Clinical history of amblyopia for this patient could not be obtained.

## 2.2. Procedure

The study protocol was reviewed and received ethics clearance through the University of Waterloo Research Ethics Committee (ORE #40773; approved 22 February 2019). Written consent was obtained from all parents or legal guardians, and all children provided a verbal assent to participate in the study. However, the consent process did not include access to the clinical information regarding the children's visual function. Testing was conducted in a quiet and well-lit room. Following a quick assessment of visual acuity (Bailey-Lovie chart) and stereoacuity (Randot® Stereotest), children performed two tests to evaluate their fine motor skills: the upper limb dexterity component of the Movement Assessment Battery for Children—version 2 (MABC-2; Pearson Canada), and a kinematic assessment of hand–eye coordination using a motion capture camera. The order of these two tests was randomized. Children wore their prescription glasses during the assessments.

The MBAC-2 test was administered according to the testing manual [8]. Briefly, children were asked to perform three tasks to evaluate their upper limb dexterity. The current study included children in two age bands: middle (7–10 years) and older (11–16 years). Tasks for the middle-age band included picking up and placing pegs using the preferred and non-preferred hands, threading lace (bimanual task), and drawing a trail using a pencil held in the preferred hand. The tasks for the older group were more difficult and included picking up, turning and placing pegs on a board using the preferred and non-preferred hands, building a triangular model (bimanual task), and drawing a trail with a pencil held in the preferred hand. All tasks were scored using criteria defined in the manual. The time required for this test was approximately 15 min.

The second assessment approach involved a bead-threading task which consisted of reaching, precision-grasping, and placement of a small object (i.e., bead diameter: 1.0 cm; hole diameter: 0.5 cm). Specifically, the children were instructed to start by placing their preferred hand in a standardized position, pinching a vertical needle (height: 12 cm; diameter: 0.3 cm) located 15 cm in front of their body's midline (see Figure 1A). The children were not restricted by a chinrest; however, the experimenter ensured that the starting position and reaching distance were constant across the trials and testing sessions. Following a "go" signal, they were instructed to quickly and accurately reach and grasp a bead, which was placed on a holder 20 cm in front of the start position, and then transport and place the bead on the vertical needle. The hand movement was recorded using a low-cost Leap motion capture (LMC) system, which records 3D hand kinematics using two cameras and three infrared LEDs. The kinematic data were recorded using a custom Java application using the Leap Motion SDK (Core Assets 4.1.1) (Lenovo, Waterloo, Canada)running on a ThinkPad T430 Lenovo laptop (Intel Core i5-3230M Processor—3.20 GHz, 3 MB Cache, 1600 MHz) (Leap Motion, San Francisco, USA). The children performed at least 5 practice trials, which were followed by 10–15 experimental trials for a total task duration of 5 min.

## 2.3. Analyses

Raw data from the MABC-2 tasks were converted to standard scores for each task and a total percentile score. These values were determined using the information/formula from the manual. The kinematic data were analysed offline using a custom MATLAB script following previously established signal processing techniques [12,15]. Each bead-threading trial consisted of a sequence with four components: reach to bead, grasp and pick up bead, reach to needle, and place the bead on a needle (Figure 1B). The peak velocity of both reaching movements (i.e., to bead and to needle) was detected, which was defined as the maximum velocity value along the depth direction. The four components were defined using the following velocity criteria: reach initiation was detected when the finger velocity exceeded 0.030 m/s for 20 consecutive ms (4 samples), and reach termination was detected when the index finger velocity dropped below 0.100 m/s for 20 consecutive ms. Reach-to-bead and reach-to-needle duration was defined as the interval between reach initiation and termination. Grasp duration was defined as the interval between reach-to-bead termination

and reach initiation of the subsequent movement. Bead placement duration was defined as the interval between reach-to-needle termination and when the hand moved away from that location indicating that the children were finished with the task.

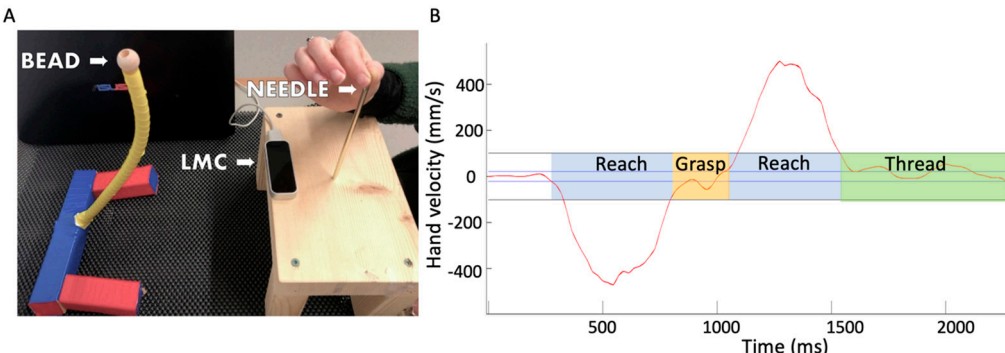

**Figure 1.** Experimental set up illustrating the position of the bead apparatus and the position of the Leap motion controller (LMC) (**A**). A typical hand velocity profile from a single trial, which consisted of four movement components: a reach to the bead (blue shaded area), grasping the bead (orange shaded area), reach to the needle (blue shaded area), and threading the bead on the needle (green shaded area). The horizontal lines represent the criteria for reach initiation (30 mm/s) and termination (100 mm/s) (**B**).

## 3. Results

### 3.1. Typically Developing Children

Table 2 shows the average results for typically developing children, summarizing their visual and motor outcomes separately for the two age bands. As expected, the cohort of typically developing children had normal visual acuity and stereoacuity. Binocular visual acuity was slightly but significantly better in the older age group compared to the younger group of children ($p < 0.05$). The administration of the MABC-2 test took approximately 15 min. On average, children in both groups scored below the 50th percentile on the manual dexterity test. Specifically, in comparison with the normative data, the average performance of the children in the middle age band was 23% lower, and the older group scored 10% lower than expected. Comparing the groups on each task revealed significantly higher standardized scores in the older compared to the younger group on the trail drawing task. The kinematic test took approximately 5 min and was completed by all children. Results from the kinematic assessment of the bead-threading revealed significant age-related improvement in the placement component, which contributed to an overall shorter movement duration (Figure 2A) in the older group compared to the younger group. Figure 2B shows the duration of each component of the bead-threading sequence for children in the two age bands. Also shown is the average performance of adults in this task taken from a previous publication [12] to illustrate the developmental trends.

Finally, a correlation analysis was conducted to determine whether the performance in the bead-threading task was associated with any of the MABC-2 upper limb dexterity scores. The results showed a significant correlation between the total movement time on the bead-threading task and the peg-placing task performed with the preferred hand, r (46) = −0.34, 95% CI [−0.06, −0.57, $p = 0.017$], and non-preferred hand, r (46) = −0.38, 95% CI [−0.11, −0.60, $p = 0.008$]. There were no significant associations between bead-threading and the other MABC-2 tasks ($p > 0.15$). To further probe the association between the bead-threading and peg-placing tasks, a regression analysis was conducted to determine which of the sequence components (i.e., reach, grasp, or place duration) explained the variance in the peg board scores. The results revealed that placement duration was the only significant variable explaining 13.5% of the variance ($R^2 = 0.135$, F(1,45) = 7.04; $\beta = -0.007$, $p = 0.011$) in the peg board task performance.

**Table 2.** Results from the visual and motor assessments stratified by age band.

| Vision and Motor Outcomes | Middle Age Band (7–10 Years Old) (n = 23, 44% Girls) | Older Age Band (11–16 Years Old) (n = 24, 46% Girls) | *t*-Test (df = 45) *p*-Value |
|---|---|---|---|
| **Vision assessment** (mean ± standard deviation) | | | |
| Binocular visual acuity (logMAR) | $-0.04 \pm 0.10$ | $-0.11 \pm 0.11$ | 2.21 $p = 0.032$ |
| Monocular visual acuity (logMAR) | $0.00 \pm 0.10$ | $-0.03 \pm 0.09$ | 1.32 $p = 0.192$ |
| Stereoacuity (arc sec) | $39 \pm 21$ | $36 \pm 27$ | 0.46 $p = 0.645$ |
| **MABC-2 assessment** (mean ± standard deviation) | | | |
| Peg-board task with the preferred hand (standard score) | $7.61 \pm 2.90$ | $9.25 \pm 3.43$ | 1.32 $p = 0.193$ |
| Peg-board task with the non-preferred hand (standard score) | $7.82 \pm 3.01$ | $9.04 \pm 3.29$ | 1.77 $p = 0.084$ |
| Manipulation task (standard score) | $8.08 \pm 3.50$ | $7.17 \pm 3.98$ | 0.84 $p = 0.406$ |
| Trail-making task (standard score) | $6.78 \pm 3.94$ | $10.37 \pm 3.10$ | 3.48 $p = 0.001$ |
| **PERCENTILE** | $27.52 \pm 27.25$ | $40.54 \pm 26.58$ | 1.66 $p = 0.104$ |
| **Kinematic assessment** (mean ± standard deviation) | | | |
| Peak velocity (m/s) | $1.183 \pm 0.154$ | $1.088 \pm 0.239$ | $t = 1.60$ $p = 0.116$ |
| Reach duration (ms) | $410 \pm 50$ | $405 \pm 96$ | $t = 0.32$ $p = 0.753$ |
| Grasp duration (ms) | $240 \pm 94$ | $192 \pm 152$ | $t = 1.30$ $p = 0.200$ |
| Thread duration (ms) | $615 \pm 169$ | $445 \pm 95$ | $t = 4.26$ $p < 0.001$ |
| **Total movement time (ms)** | $1675 \pm 282$ | $1452 \pm 275$ | $t = 2.74$ $p = 0.008$ |

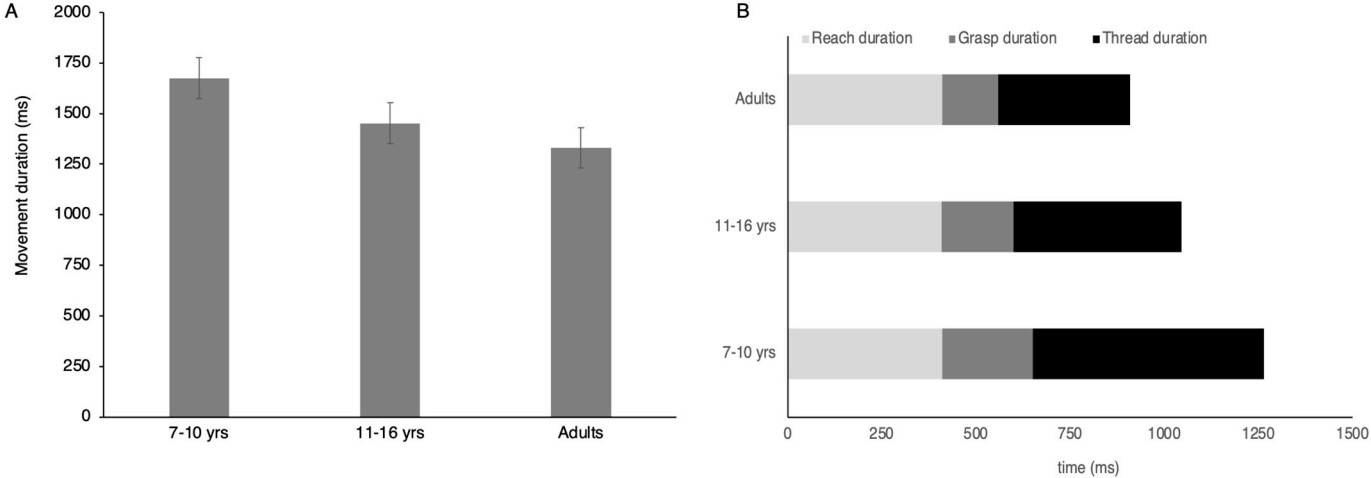

**Figure 2.** The average total movement duration for the bead-threading task plotted for the two groups of children and adults (**A**). The average duration of each task component (**B**) illustrating the effects of age is most pronounced for the grasping and threading.

*3.2. Clinical Application—Case Study*

The tests on fine motor skills provided convergent results for the two children with a history of amblyopia (Figure 3). Patient 1 performed slightly below average, achieving a standardized MABC-2 score at the 16% percentile. In comparison to the age-matched

control group, this child also exhibited a 20% increase (i.e., 120 ms) in the duration of the reaching and grasping components of the bead-threading task; however, the threading duration was similar to the control group. In contrast, the standardized MABC-2 score for Patient 2 fell below the first percentile, which indicates very poor fine motor skills. Analogous results were revealed through the kinematic assessment, where all aspects of prehension were prolonged. The greatest deficit was seen for the object manipulation components where grasping and threading durations were each 1000 ms longer, suggesting a major deficit in fine manipulation skills.

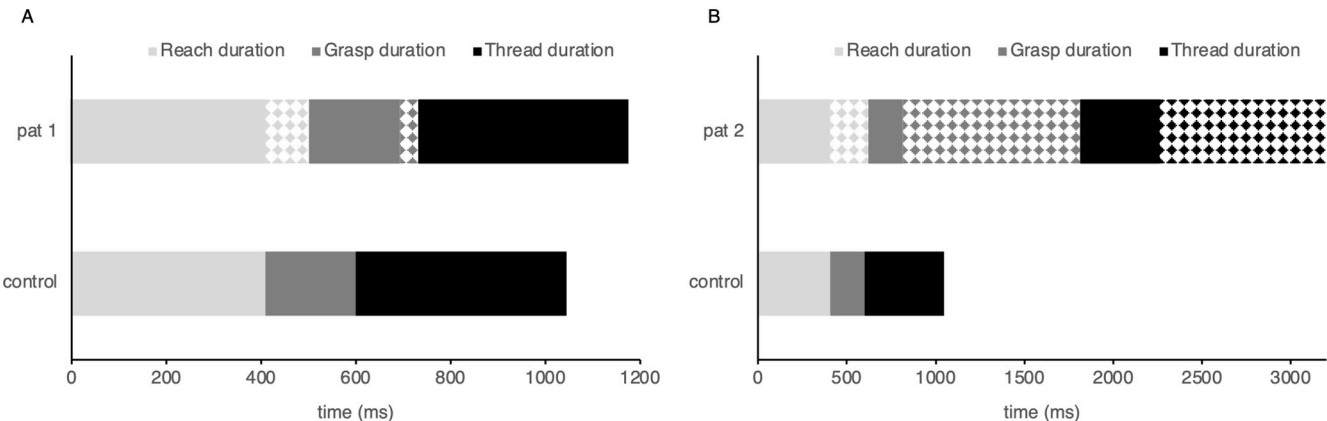

**Figure 3.** Duration of reaching, grasping, and threading plotted for patient 1 (**A**) and patient 2 (**B**). The solid colors represent the duration found in the age-matched control group; the hatched area highlights the changes in duration of each component illustrating where patients are spending more time in comparison to the control group.

## 4. Discussion

Eye–hand coordination and fine motor skills are important for most daily activities. Evaluating these skills, developing rehabilitation regimens to address deficits, and monitoring the effects of therapy are necessary for the development of proper visuomotor control. Vision provides important sensory input for the planning and execution of fine motor skills; thus, unsurprisingly, children with visual impairments due to amblyopia or strabismus often experience deficits in visuomotor coordination. Assessing such deficits during the course of treatment could provide important insight into the effects of therapy on everyday functions. The aim of this study was to compare two approaches that can be used to assess upper limb motor performance: a standardized clinical test and a technology-based kinematic approach. Our results indicate that these approaches provide converging results. Importantly, using a kinematic assessment is relatively quicker and offers additional insights into visuomotor control, which cannot be gleaned from the clinical tests alone.

The main advantage of using a standardized clinical test is that it allows for a comparison of the child's score with previously established norms based on very large cohorts. Such scores can be useful to determine if the child's performance is within age-expected norms or falls below norms, indicating a potential deficit in fine motor skills. The cohort tested in the current study consisted of typically developing children without a medical diagnosis of ADHD, learning disability, or autism; however, the mean standardized scores in both age bands fell below the 50th percentile. This unexpected shift towards a poorer performance was also noted in a study by Gaul and Issartel [16]. Although the underlying reasons for the shift in motor proficiency are not well understood, it has been suggested that children today are more engaged with technology (i.e., typing and touching screens on tablets and phones) and less engaged with activities involving fine motor skills (i.e., writing, cutting with scissors, building construction blocks, or solving jigsaw puzzles). The emerging trend towards lower performance on clinical tests found in the current study may pose some difficulty when interpreting the standardized scores.

Historically, fine motor skills have been assessed using a variety of tasks by measuring the time it takes to accomplish the task and observing errors [17,18]. However, such approaches are time-consuming. For example, the administration of the upper limb dexterity test in the current study took approximately 15 min. In contrast, the kinematic assessment of the bead-threading task took approximately 5 min. The main factor contributing to this difference was the number of tasks used to evaluate performance. The clinical score for the upper limb dexterity test was based on the performance scores across four tasks, whereas the kinematic assessment involved a single task. Thus, it is important to ask whether a single task can provide sufficient information regarding motor skill proficiency. The answer to this question depends on the purpose of the assessment.

A myriad of tasks could be used to assess fine motor skills. The tasks used in the MABC include a discrete sequencing task (i.e., the peg board), a bimanual coordination task (i.e., stringing beads or making a model), or a continuous tracing task (i.e., drawing a trail). Performance on these tasks relies on different sensorimotor processes; thus, it is not surprising that test scores were not correlated significantly with each other. On the other hand, bead-threading was the single task used for the kinematic assessment, and it can be described as a discrete sequencing task. Performance on the bead-threading task was moderately yet significantly correlated with the scores from the peg board task, suggesting that both tasks engage similar visuomotor processes involved in reaching, grasping, and object placement. Current results are in line with previous work where moderate correlations among different clinical tests of motor proficiency have been reported [19]. The advantage of using a kinematic assessment is the unique insight into the individual prehension components and the coordination between the reaching and object manipulation components.

Importantly, recent research has begun to reveal the contribution of binocular vision to the control of prehension during the performance of a bead-threading task [20]. For example, in a cohort of typically developing children and adolescents, those with a higher level of sensory binocularity or stereoacuity were found to have a more efficient grasp execution. Previous work has also revealed an association between horizontal fusional reserves with a higher reach peak velocity, as well as an association between accommodative function and the duration of placing the bead on the needle [21]. Analogous findings have been reported in children with amblyopia and/or strabismus, where reduced stereoacuity has been associated with longer grasping and object placement [22]. Interestingly, the reach component seemed less affected in that cohort. The two case studies presented in this paper are in line with previous reports [2,23,24] and highlight the spectrum of deficits in visuomotor control in children with amblyopia, which could range from relatively mild to very severe cases where all aspects of visuomotor control seem to be affected. Importantly, the current study indicates that impairments are detected through both the clinical and kinematic approaches, suggesting that a quick kinematic assessment may be sufficient to diagnose poor visuomotor control. Notably, having some information about the execution and coordination of the prehension components (i.e., reach and grasp) based on the kinematic output could be helpful. For example, it is possible that a different treatment approach may be required in cases where a poor performance is due to a complete disruption of visuomotor control in comparison to cases where a mild disruption in skill proficiency may be due to a less efficient grasping or object manipulation. Moreover, a detailed kinematic assessment might have a higher specificity and sensitivity at detecting small but meaningful changes in visuomotor control [25–28]. For example, management plans that aim to improve stereoacuity might have a significant impact on grasp execution. Finally, a kinematic approach might be advantageous for monitoring the effects of therapy because changes can be documented using the same task(s) in children and adolescents with no ceiling effect. In contrast, clinical tests often use different tasks across age bands; thus, in some cases when the child's treatment extends over several months or years, they might be tested with different tasks.

Advancements in technology have led to the development of portable and user-friendly systems and devices that could modernize the clinical assessment of fine motor skills [29–31]. However, the development and implementation of kinematic assessments for clinical purposes will require significantly more research [32] to overcome the many challenges that limit the practical application within a clinical setting. First, the technology allows for the seamless recording of movement kinematics; however, the results are not immediately available as the analyses of such datasets remains time-consuming. With recent advances in machine learning and the application of artificial intelligence [33], these challenges are likely to be addressed in the near future. The second challenge, and the main limitation of the current work, is the exploration of a single bead-threading task. Work in our lab strongly supports that this task provides a sensitive behavioural assay into visuomotor control and its maturation [13], although other tasks must be developed to provide complementary insight into a full gamut of fine motor skills. The development of such tasks should be guided by knowledge of the maturation trajectory of visuomotor control [34–38].

This is the first study to directly compare the results from a standardized clinical test of fine motor skills (i.e., MABC) with a technology-based assessment of prehension. This is an important first step to move towards applying technology outside the research lab in real-world settings. Our results suggest the potential utility of this approach; however, the study has several limitations. For example, the protocol did not include a full optometric exam, which may help to explain the poorer performance in some children. Although the children were recruited from the optometry database and assessed in the optometric setting, their full medical history was not available to the researchers. The shift towards poorer fine motor skill performance was not expected in the cohort of typically developing children, where acuity and stereoacuity were within a normal range. Such narrow range of results from vision tests also precluded a correlation analysis between motor performance scores and visual acuity or stereoacuity. Nonetheless, it is possible that poorer motor performance scores are associated with aspects of binocular function not assessed in the current study, such as vergence or accommodation, which is common in children [39–41] and young adults [42–44]. Future studies should consider adding a full optometric exam to the study protocol.

## 5. Conclusions

This study investigated the feasibility of using a kinematic assessment of fine motor skills within an optometric setting. The results support that in comparison to a clinical test, a kinematic assessment provides converging results and additional insights into visuomotor control. Adding a kinematic assessment could reveal aspects of visuomotor control that may be difficult to quantify objectively using the current clinical approaches, which are based on the observation of performance. Specifically, the bead-threading task provides a detailed evaluation of the components of a prehension task where the reach duration and object manipulation such as grasping and placement can be assessed objectively and quantified precisely. Such information could be helpful by providing insights into which part of the prehension task may be contributing to lower scores in motor performance and may be useful in diagnosing and managing treatment outcomes. For example, the bead-threading task appears to depend on binocular vision; thus, it might be useful for monitoring the functional impact of new amblyopia therapies targeting the recovery of binocular vision. Therefore, the use of technology could advance and modernize the clinical approaches for fine motor skill assessment; however, further research is required to examine the reliability, validity, and implementation across a wider range of clinical settings. Finally, given that vision, eye movements, and upper limb movements are tightly linked, the assessment of these systems should be integrated and requires an interdisciplinary approach where behavioural optometrists and allied rehabilitation providers come together to address visual and movement difficulties.

**Author Contributions:** Conceptualization, E.N.-S., T.A.B., B.T. and L.W.T.C.; methodology, E.N.-S., T.A.B., B.T. and L.W.T.C.; formal analysis, E.N.-S.; investigation, E.N.-S.; resources, E.N.-S., B.T. and L.W.T.C.; data curation, E.N.-S.; writing—original draft preparation, E.N.-S.; writing—review and editing, E.N.-S., T.A.B., B.T. and L.W.T.C.; project administration, E.N.-S.; funding acquisition, E.N.-S. and B.T. All authors have read and agreed to the published version of the manuscript.

**Funding:** This research was funded by ENS NSERC grant RGPIN-03148. B.T. is supported by InnoHK and the Hong Kong SAR government, NSERC grants RPIN-05394 and RGPAS-477166 and CIHR grant 390283.

**Institutional Review Board Statement:** The study protocol was reviewed and received ethics clearance through the University of Waterloo Research Ethics Committee (ORE #40773).

**Informed Consent Statement:** Written consent was obtained from all parents or legal guardians, and all children provided a verbal assent.

**Data Availability Statement:** All data are available upon reasonable request.

**Conflicts of Interest:** The authors declare no conflicts of interest.

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
