# Peer review of "Kinematic Assessment of Fine Motor Skills in Children: Comparison of a Kinematic Approach and a Standardized Test"

_2411-5150, 2023_

Round 1

Reviewer 1 Report

Comments and Suggestions for Authors

The authors explored kinematic assessments within an optometric setting with typically developing children with normal vision, and two amblyopic children. Children with normal vision were found to have poorer upper limb dexterity than expected. The authors suggested that the increased use of technology today may provide less opportunities in developing fine motor skills. While this may be true, it is more likely that a subset of participants has undiagnosed binocular dysfunctions and uncorrected astigmatism. For example, the prevalence of binocular dysfunction was previously found to be more than 30% among university students in the absence of strabismus, amblyopia and/or any significant uncorrected refractive error (e.g. Porcar and Martinez-Palomera, 1997). Therefore, the authors should report if such a trend is observed in this cohort. In addition, the authors should provide information on how they determine the children are typically developing. The authors reported deficits in fine motor control with the two amblyopic children but they did not mention the type of amblyopia, the magnitude of refractive errors, their treatment history and if spectacles were used during the assessment. Since the research is conducted in an optometric setting, the authors should also briefly mention the role of behavioral optometrists in the treatment of these conditions. Overall, the article is well written and should be published once these comments are addressed.

Author Response

Thank you for reviewing the paper and positive feedback. We agree that difficulties with motor skill performance may arise to do undiagnosed binocular dysfunctions or uncorrected astigmatism. This is an important topic for research, unfortunately, a detailed optometric assessment was not performed with the children who participated in this study. The parameters assessed included only binocular and monocular acuity, and stereoacuity. Similarly, we were not able to access a detailed exam for the two case studies of children with reduced acuity. Therefore, we cannot look at the associations suggested by the reviewer. A parallel study currently underway in our lab is collecting more detailed information. We would like to reiterate that the goal of the study described in this submission was to determine whether technology could provide useful information about motor function with a relatively quick assessment. We have acknowledged the limitations of not being able to report this information (line 316). Despite this limitation, our results suggest that children may benefit from this type of assessment, which was the main purpose of the experiment.

Children were recruited form an Optometry Clinic database, and any children with diagnosed neurodevelopmental disorders, such as ADHD, learning disability or autism were excluded using a screening questionnaire. Thus, typically developing children were defined as a children without a medical diagnosis of these disorders. Parents were asked to ensure that children who wore prescription glasses brought their glasses for the study appointment. The accumulating research suggests that visual function, in particular binocular vision, provides important sensory input for motor skill development. Therefore, behavioural optometrists may have an important role in monitoring the development of motor skills, and assessing the effects of visual therapies. We addressed all these points in the revised manuscript.

Reviewer 2 Report

Comments and Suggestions for Authors

The study “Kinematic Assessment of Fine Motor Skills in Children in a 2 Clinical Optometric Setting” examines fine motor skills in children, especially those with conditions like amblyopia. The main goal is to test the practicality of a kinematic assessment in an optometric setting using affordable equipment and to see if it adds value to a standard motor function test (MABC). The research involved 47 typically developing children, assessing upper limb dexterity with MABC-2 and recording hand movements with the LEAP motion capture system. Even though visual abilities were normal, MABC-2 scores were lower than expected. Comparing MABC-2 and kinematic measures in two children with amblyopia revealed consistent findings, highlighting fine motor control deficits. In conclusion, kinematic assessment proves beneficial in an optometric context, complementing standard tests for measuring visuomotor skills.

I think the study is interesting because it aimed to assess the practicality of using a portable motion capture device in a clinical optometric environment to evaluate fine motor skills. It compared results from a kinematic assessment with a standard fine motor skills clinical test to determine the viability of this technology-based approach in an optometric setting. The research addressed limitations in current standardized clinical tools, such as MABC-2 and BOT-2, which fail to pinpoint disruptions in visuomotor control underlying poor performance. The study proposed using high-speed motion cameras for detailed kinematic measurements to complement clinical assessments and monitor treatment effects. By identifying compensatory kinematic strategies and understanding sensorimotor adaptations in children, the study aimed to facilitate early intervention for those at risk of poor visuomotor development. The exploration utilized new, inexpensive, and compact technology, making in-clinic kinematic assessment feasible for children with normal vision and those with amblyopia.

But the optometric part needs to be reinforced, it is said that it is like one of the objectives but little is taken into account the optometric examination or binocular vision values. I recommend making the following corrections:

Abstact:

Define LEAP:

Introduction:

The introduction is well presented, although I would recommend grouping the objective of the study together at the end, which seems to be divided between paragraphs 1 and 2.

Methods:

What was the refraction of each eye of the two amblyopic subjects?

Was the refraction of each eye of each participant taken into account?

What type of amblyopia did each patient suffer from?

Did the children who wore glasses perform the tasks with their usual correctness? What were the inclusion criteria for the study in terms of visual acuity and refraction?

In addition to measuring visual acuity, was a complete optometric examination performed? This aspect seems crucial to me, especially given the incidence of accommodative and vergential dysfunctions that are occurring at this time, whether due to the use of screens, or staying indoors for prolonged periods of time due to the Covid-19 Pandemic. I also think it is very important to know if any of these children had myopia or pseudomyopia.

I think the authors should give more details of the optmetric protocol or justify if it was not taken into account and be aware of the limitations and biases that not considering all of this can bring to the study.

Another important aspect is whether it was considered that the working distance when performing these tasks was always the same for all subjects, since this directly influences the results, and it is not explained if it was controlled to be constant all the time, such as using a chin rest or something that would not cause the participants to change the indicated distance.

Results:
As the authors discuss, people with amblyopia or strabismus often experience deficits in visuomotor coordination, so it is crucial to address the motor problem having performed a complete optometric examination, thus being able to prescribe a complete comprehensive rehabilitation. I think they should emphasize optometric values to correlate them with deficits in fine motor skills.

The discussion is well stated.

Author Response

Comment 1: Abstract: Define LEAP:

Author reply: Leap is the name of a commercially available device, it is not an acronym. We changed the spelling to lowercase to improve clarity.

Comment 2: The introduction is well presented, although I would recommend grouping the objective of the study together at the end, which seems to be divided between paragraphs 1 and 2.

Author reply: Thank you, the objectives are described in the first paragraph in the revised manuscript.

Comment 3: What was the refraction of each eye of the two amblyopic subjects?

Author reply:  Unfortunately, a detailed optometric assessment was not performed with the children who participated in this study. The parameters assessed included only binocular and monocular acuity, and stereoacuity. Similarly, we do not have a detailed exam for the two case studies of children with reduced acuity.

Comment 4: Was the refraction of each eye of each participant taken into account?

Author reply: The refractive error was not measured as part of the protocol in this study

Comment 5: What type of amblyopia did each patient suffer from?

Author reply: One participant had a history of strabismic amblyopia, which was confirmed by a clinical history of strabismus surgery and patching based on parental report. The clinical details for the other participant cannot be accessed by the research team. This information was added to the table. We fully agree that the lack of this clinical information is a big limitation of the current paper, however, the study’s main objective was to assess the utility of technology. Given the importance of clinical information, we have reached out to the family but have not received a response.

Comment 6: Did the children who wore glasses perform the tasks with their usual correctness? What were the inclusion criteria for the study in terms of visual acuity and refraction?

Author reply: All children were asked to wear their regular prescription glasses for the study appointment (added on line 103 in the revised manuscript). The study used a sample of convenience where visual acuity and refraction were not part of the inclusion criteria, any children with diagnosed neurodevelopmental disorders, such as ADHD, learning disability or autism were excluded using a screening questionnaire.

Comment 7: In addition to measuring visual acuity, was a complete optometric examination performed? This aspect seems crucial to me, especially given the incidence of accommodative and vergential dysfunctions that are occurring at this time, whether due to the use of screens, or staying indoors for prolonged periods of time due to the Covid-19 Pandemic. I also think it is very important to know if any of these children had myopia or pseudomyopia.

Author reply:  A complete optometric examination was not performed in this study, which is a limitation of this research as difficulties with motor skill performance may arise to do undiagnosed binocular dysfunctions. We discuss this limitation in the revised manuscript (line 311-325). We would like to reiterate that the goal of the study described in this submission was to determine whether technology could provide useful information about motor performance with a relatively quick assessment.

Comment 8: I think the authors should give more details of the optmetric protocol or justify if it was not taken into account and be aware of the limitations and biases that not considering all of this can bring to the study.

Author reply: Our paper was revised to include a discussion addressing the lack of a detailed visual assessment. Please see Lines 311-325.

Comment 9: Another important aspect is whether it was considered that the working distance when performing these tasks was always the same for all subjects, since this directly influences the results, and it is not explained if it was controlled to be constant all the time, such as using a chin rest or something that would not cause the participants to change the indicated distance.

Author reply: A chinrest was not used in this study for the kinematic assessment and it is not part of the standardized clinical test of motor skills (MABC). The experimenter ensured that the apparatus and equipment were always placed in the same location and the child was asked to maintain a stable trunk position, which was monitored by the experimenter. It is possible that the child leaned forward by a small amount which would change the vergence eye movements. The distance between the hand start position and the bead was kept constant, thus, if the child leaned forwards slightly it would not influence the hand kinematic results because the reaching distance was fixed. This information was added in the methods section (Line 119)

Comment 10: As the authors discuss, people with amblyopia or strabismus often experience deficits in visuomotor coordination, so it is crucial to address the motor problem having performed a complete optometric examination, thus being able to prescribe a complete comprehensive rehabilitation. I think they should emphasize optometric values to correlate them with deficits in fine motor skills.

Author reply:  Accumulating research supports that people with amblyopia and strabismus are likely to have difficulties with motor skill performance. Thus, understanding the association between vision and motor control is quite important. The current study included typically developing children with normal visual acuity and stereoacuity which limits the range of values measured. Given the narrow distribution of the visual measures and a relatively small sample size, a correlation analysis might not provide reliable information. This limitation was added in the revised manuscript in the Discussion section (line 320)

Round 2

Reviewer 2 Report

Comments and Suggestions for Authors

Dear authors,

I think the responses are partially done.

The title of the article highlights that the assessment took place "in a clinical optometric setting"; however, there is a notable discrepancy as no optometric evaluations, including refraction or assessment of binocular abilities, were conducted. Only visual acuity (VA) and stereoacuity were measured. I believe there is a fundamental issue in not incorporating a comprehensive optometric evaluation for each patient. Consequently, characterizing the study as an optometric one may be misleading, as a well-executed optometric study should adhere to established parameters widely recognized in the field to avoid potential biases in the obtained results.

On another note, I recommend retaining the complete study objective at the end of the introduction rather than condensing it all into the first paragraph. This approach will provide a clearer structure and enhance the overall coherence of the study.

Author Response

We fully agree that a major limitation of this study is the lack of a full clinical assessment of visual functions. Thus, the title may have been misleading. We would like to change the title to: "Kinematic Assessment of Fine Motor Skills in Children: Comparison of a Kinematic Approach and a Standardized Test".
The methods section was also revised to include a statement indicating that the consent process did not involve disclosure of clinical information regrading children's visual function. Finally, the objective was moved to the last paragraph of the introduction section.